# Making Video Models Adhere to User Intent with Minor Adjustments

**Daniel Ajisafe**                                           *dajisafe@cs.ubc.ca*
*Department of Computer Science*
*The University of British Columbia*

**Eric Hedlin**                                              *iamerich@cs.ubc.ca*
*Department of Computer Science*
*The University of British Columbia*

**Helge Rhodin**                                 *helge.rhodin@uni-bielefeld.de*
*Faculty of Technology*
*Bielefeld University*
*Department of Computer Science*
*The University of British Columbia*

**Kwang Moo Yi**                                                  *kmyi@cs.ubc.ca*
*Department of Computer Science*
*The University of British Columbia*

**Reviewed on OpenReview:** *https://openreview.net/forum?id=Opvq2wfBR5*

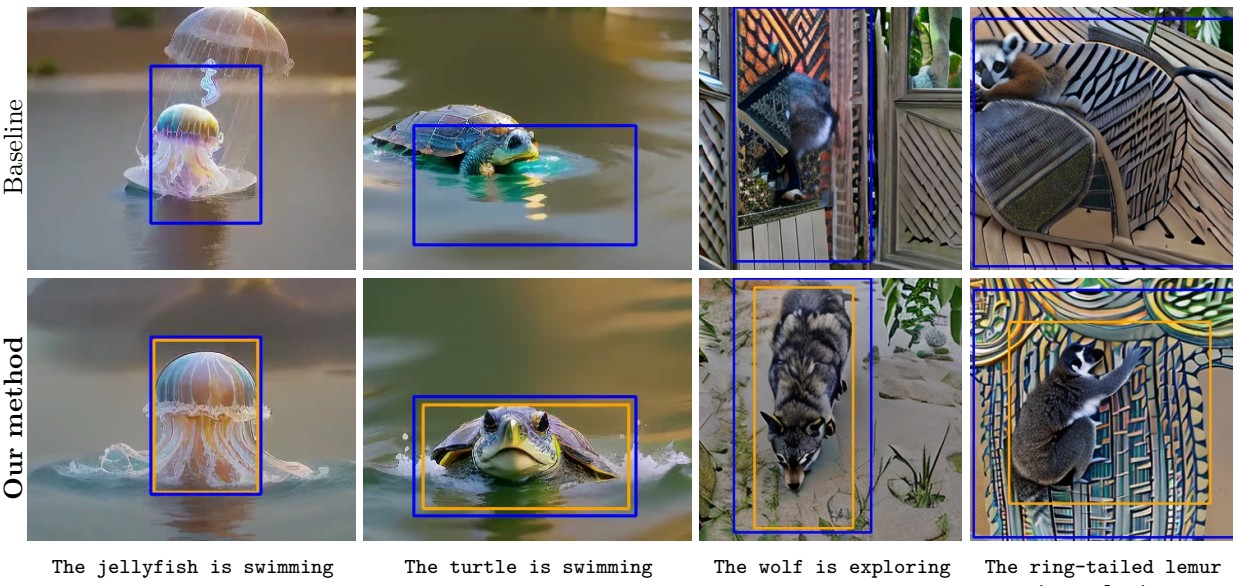

The jellyfish is swimming     The turtle is swimming     The wolf is exploring     The ring-tailed lemur is exploring

Figure 1: **Teaser** − We show example bounding box controlled video generations with (left) Trailblazer Ma et al. (2024b) in T2V-Turbo Li et al. (2025) and (right) the original Trailblazer Ma et al. (2024b) respectively. On top is the original control signal and below is our adjusted bounding boxes. We show the original bounding boxes as blue and the adjusted boxes as orange. While the modification is subtle, the difference in the quality of generation is large. We modify bounding boxes to adhere better to the cross-attention maps within the video models.

## Abstract

With the recent drastic advancements in text-to-video diffusion models, controlling their generations has drawn interest. A popular way for control is through bounding boxes or layouts. However, enforcing adherence to these control inputs is still an open problem. In this work, we show that by slightly adjusting user-provided bounding boxes we can improve both the quality of generations and the adherence to the control inputs. This is achieved by simply optimizing the bounding boxes to better align with the internal attention maps of the video diffusion model while carefully balancing the focus on foreground and background. In a sense, we are modifying the bounding boxes to be at places where the model is familiar with. Surprisingly, we find that even with small modifications, the quality of generations can vary significantly. To do so, we propose a smooth mask to make the bounding box position differentiable and an attention-maximization objective that we use to alter the bounding boxes. We conduct thorough experiments, including a user study to validate the effectiveness of our method. Our code is made available on the project webpage to foster future research from the community.

## 1 Introduction

Text-to-video diffusion models have made groundbreaking advances in producing high-quality prompt-directed generations Ho et al. (2022b); Singer et al. (2023); Weissenborn et al. (2020); Arnab et al. (2021); Ho et al. (2022a); Chen et al. (2024a). Among the various directions, methods based on diffusion Ho et al. (2022b;a); Chen et al. (2024a) and transformers Weissenborn et al. (2020); Arnab et al. (2021) have become popular. While these models can be controlled through proper prompting, this is not always straightforward and requires careful prompt engineering Liu et al. (2023). In particular, the spatial control of object placement and object trajectories remains difficult.

Naturally, researchers have sought to improve the controllability of text-to-video diffusion models. These include methods that specifically train a model in addition to the main model Zhang et al. (2023), which was shown to be especially effective for text-to-image generation Wang et al. (2023); Zhang et al. (2023); Patashnik et al. (2021). For video generation, however, training such additional models is computationally expensive. For spatial control, using bounding boxes or layouts as control inputs Ma et al. (2024b); Zheng et al. (2023) has gained attention. These approaches work without additional training by simply modifying the internal attention maps within the video diffusion model Ma et al. (2024b); Hertz et al. (2023); Chefer et al. (2023); Tumanyan et al. (2023) or through guidance Patashnik et al. (2021); Nichol et al. (2022) with a pre-trained classifier. While these methods are effective, they are still limited in terms of adherence to the control inputs. Generated videos contain artificial outcomes because of the mismatch between how the control signals affect the video generation process and how it was trained without such injection; see Figure 1.

In this work, we show that by slightly adjusting the control inputs, we can improve both the quality of generations and the control adherence. However, finding how to adjust bounding boxes to adhere to user control without hurting the final generation outcome is challenging. For instance, modifying and balancing attention within the diffusion model is effective for control but can lead to over-saturation. In general, changes that are outside of the model's internal understanding easily cause video generation to degrade. To address these, we optimize bounding boxes to better align with the internal attention maps of the video diffusion model, while still being close to the user-provided bounding boxes. More specifically, we introduce an optimization framework that ensures that attention maps edits remain differentiable with respect to the bounding box parameters. With the differentiable pipeline, for improved alignment of the bounding box control and the video model, we then propose to maximize the attention within the bounding box of the *next layer after the edit*, which is representative of where the neural network is focusing on once the edits are applied. Further, while we enhance focus, we also balance attention between the foreground and background areas, so that generation process does not completely ignore the background. While the resulting adjustments to the bounding boxes are small, their impact on the video generation is significant; see Figure 1.

To summarize, we make the following contributions:

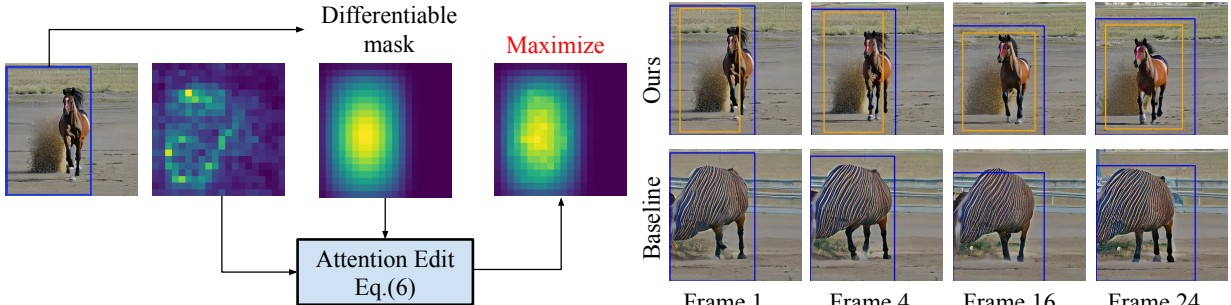

Figure 2: **Overview** – We inject bounding box control for video diffusion models by editing their cross attention maps within the network. However, not all such edits are friendly to video diffusion models as they are not trained with such edits. Thus, when applying these edits, we make sure that this editing process is differentiable (Section 3.2) and adjust the edit parameters in a way such that the network behaves as intended—attention being focused on desired regions (Section 3.3). We show the original bounding boxes in blue and the adjusted bounding boxes in orange. Though the adjustments are minimal and close to the original user input, they create a drastic difference in terms of video generation quality and adherence to the bounding boxes.

- we demonstrate that small adjustments to user intent can lead to significant improvements in controlled video generation when adjusting with our method;
- we propose a novel method to optimize the control inputs to align well with the internal attention maps of the video diffusion model, while still being close to the user-provided bounding boxes;
- to do so, we present an editing pipeline that alters a non-differentiable method to be differentiable with respect to the bounding box parameters;
- to find better bounding-box parameters, we propose a balanced attention maximization objective considering the cross-attention maps of the next layer after the edits; and
- we conduct experiments including a user study to validate the effectiveness of our method, and show that it outperforms existing methods in terms of video generation quality and adherence to user intent.

## 2 Related Work

**Text-to-image generative models.** Generative text-to-image models have shown ground-breaking results in terms of high-quality images Rombach et al. (2022); Nichol et al. (2022) synthesizing different objects, persons, and places. However, these models have been shown to fail to adhere faithfully to spatial user intent Rombach et al. (2022); Ramesh et al. (2022); Nichol et al. (2022), hence requiring a separate modality e.g., boxes, scribble, etc., for better text-to-image alignment.

**Text-to-image generative models with box control.** With alternate input control such as simple user-defined boxes, several methods Zheng et al. (2023); Chang et al. (2023) have demonstrated that the spatial composition can be controlled more faithfully. For example, Directed diffusion Ma et al. (2024a) directs the placements of objects by introducing activations at desired positions in a text-to-image diffusion model, leading to a better generation outcome. BoxDiff Xie et al. (2023) introduces spatial constraints such as inner-box, outer-box, and corner constraints in an optimization framework, for controlling objects and context in generated images. These methods follow a forward guidance approach. Alternatively, other work Chen et al. (2024b) demonstrates the superiority of backward guidance over forward guidance for robust layout control. Their method optimizes the latent, allowing both guided and unguided tokens to influence the generation outcome. However, these methods are limited to single images and do not investigate their effectiveness on temporal data, i.e., video.

**Text-to-video generative models with box control.** There are several works that investigate the adherence to input control for videos Wang et al. (2023); Lian et al. (2024); Jain et al. (2024); Ma et al. (2024b); Wang et al. (2024); Chen et al. (2025); Luo et al. (2025); Qiu et al. (2024); Lei et al. (2025). None, however, look into how slight changes in controls can lead to diverse final generation outcomes. Peekaboo Jain et al. (2024) uses attention masking to guide a video generation process, therefore utilizing local context for generating individual objects. However, their use of an infinite attention injection in the background often results in missing background details Ma et al. (2024b). Trailblazer Ma et al. (2024b), on the other hand, uses a direct/in-place replacement strategy to bias cross-attention maps towards user intent. With keyframed boxes, their method achieves controllable generation but often sacrifices better generation outcomes for more control. In other words, controlled generation outcomes are less faithful to the given prompt. Their method also relies on tuning hyper-parameters per object, which is not scalable. Freetraj Qiu et al. (2024) is also a training-free method that achieves controllable video generation by imposing guidance on both noise construction and attention computation. While it achieves good results, its trajectory injection sometimes leads to poor visual quality or poor trajectory alignment.

Other methods, such as Motion-Zero Chen et al. (2025) and Boximator Wang et al. (2024), do not release implementation code or evaluation data, making fair experimental comparison difficult. Ctrl-V requires task-specific training and conditions video generation on an initial frame, whereas our method is training-free and operates directly on text-to-video diffusion backbones. Generally, other methods also differ substantially in supervision regime or underlying model architecture, which limits fair comparison.

To the best of our knowledge, we are the first to explore the benefit of adjustments to box-controlled generation outcomes. We build on the work of Trailblazer Ma et al. (2024b), and apply our method to two text-to-video models Ma et al. (2024b); Li et al. (2025). We demonstrate our core contributions in the following sections.

## 3 Method

Our core idea is to modify user input, i.e., the bounding boxes, aligning them to the internal attention mechanisms within the video diffusion model. As shown in Figure 2, we implement this by introducing a differentiable attention map editing method, which we then optimize through to adjust the bounding boxes. When adjusting, as we do not have any specific measure for what a good trajectory is—after all, we are generating a video from scratch—we look into how the attention maps evolve through the network layers. Specifically, we encourage the edits to be inline with what how the neural network creates further attention maps, that is, we regularize it such that the attention map, after editing, causes the attention map of the next layer within the neural network to focus within the box.

To explain our method, and for completeness, we first review how control injection is done for a training-free baseline, then discuss our work.

### 3.1 Preliminary: Trailblazer Ma et al. (2024b)

Our goal is to generate a video with $F$ frames, where a desired object of interest, e.g., "cat", adheres faithfully to the motion and control of the bounding boxes $\mathcal{B} \in \mathbb{R}^{F \times 4}$ provided by the user. We build upon Trailblazer Ma et al. (2024b), which 'injects' user control in the form of bounding boxes, by directly adjusting the internal attention maps of the video diffusion model. A strong benefit in doing so is that there is no need to train or fine-tune the video diffusion model, and any off-the-shelf video diffusion model can be used as long as it has cross-attention layers.

Specifically, at inference, we feed an input text embedding $\boldsymbol{p}$, noisy latent code $\boldsymbol{z}$, and timestep $t$, as input to a video diffusion model $\Theta_b$. After each cross-attention layer, we extract and edit cross-attention maps $\boldsymbol{A}_S \in \mathbb{R}^{C \times H \times W \times N}$, temporal cross-attention maps $\boldsymbol{A}_T \in \mathbb{R}^{C \times H \times W \times F \times F}$. Then, denoting the predicted noise sample as $\epsilon_t$, we can write

$$\boldsymbol{A}_S, \boldsymbol{A}_T, \epsilon_t = \Theta_b(\boldsymbol{p}, \boldsymbol{z}, t), \tag{1}$$

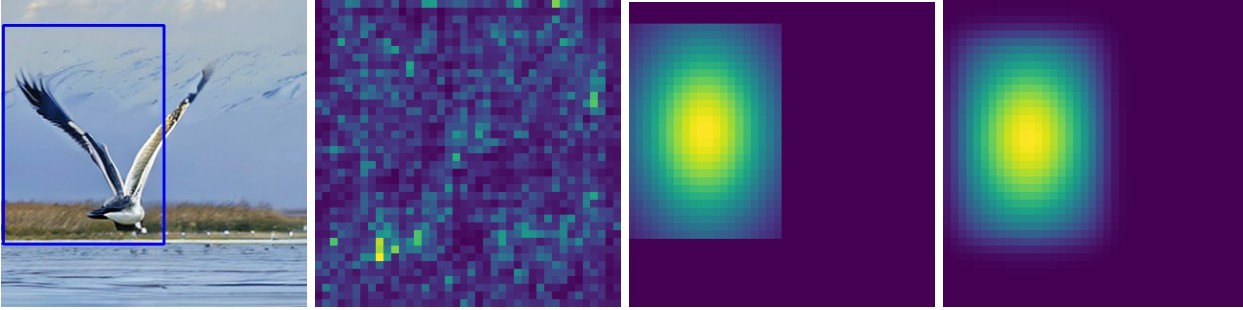

(a) Generated frame and user control (b) Attention map for the token 'Swan' (c) Attention map edited by Trailblazer Ma et al. (2024b) (d) Attention map edited by our method

Figure 3: **Example of attention map editing** – We show an example of the generated video frame with the desired user control bounding box, and the associated attention map edits. While effective, Trailblazer Ma et al. (2024b) relies on a replacement operation that is not differentiable with respect to the box parameters. Our method, on the other hand, performs a smooth differentiable edit.

where $N$ is the number of text tokens, and $N = 77$ when using CLIP tokenizer Radford et al. (2021). Here, the control signal is then injected by modifying the attention maps $\boldsymbol{A}_S$ and $\boldsymbol{A}_T$. Note here that $\boldsymbol{A}_S$ encodes the relationship between the feature $\boldsymbol{V}$ and text embedding $\boldsymbol{p}$, while $\boldsymbol{A}_T$ encodes the relationship between pixels among different frames, without any association to the text.

In more detail, the attention maps are modified by directly weakening the attention map outside of the bounding box via multiplying a constant weakening factor, and strengthening the attention map within the box by adding a Gaussian map. Consider now the masks $\boldsymbol{M}_{\mathcal{B}_S} \in \mathbb{R}^{C \times H \times W \times N} \in [0, 1]$ and $\boldsymbol{M}_{\mathcal{B}_T} \in \mathbb{R}^{C \times H \times W \times F \times F} \in [0, 1]$ for the spatial and temporal layers, respectively, which encodes whether the pixels involved in the attention maps are either within (1) the control bounding box or not (0). These maps $\boldsymbol{M}_{\mathcal{B}}$ are anchored by box parameters $\boldsymbol{b} \in (b_l, b_t, b_r, b_b) \in \mathbb{R}^4$ representing top, left, right and bottom coordinates. $C, H, W$ stands for the number of channels, the height, and the width respectively. Trailblazer Ma et al. (2024b) then modifies the attention maps as

$$\bar{\boldsymbol{A}}_* = \underbrace{\boldsymbol{A}_* \odot (\lambda_{\mathrm{w}}(1 - \boldsymbol{M}_*) + \boldsymbol{M}_*)}_{\text{weakening outside}} + \underbrace{\lambda_{\mathrm{s}} \boldsymbol{M}_{\mathcal{G}} \odot \boldsymbol{M}_*}_{\text{strengthening inside}} \quad , \tag{2}$$

where subscript $*$ denotes both spatial and temporal, $\odot$ is the element-wise multiplication, $\lambda_{\mathrm{w}}$ is the weakening factor which we set to $\lambda_{\mathrm{w}}{=}0.001$, and $\lambda_{\mathrm{s}}$ is the strengthening factor which we set to $\lambda_{\mathrm{s}}{=}0.15$,[1] and $\boldsymbol{M}_{\mathcal{G}}$ is a Gaussian map, defined as[2]

$$\boldsymbol{M}_{\mathcal{G}} = \exp\left(-\frac{(\boldsymbol{I}_{coord} - \boldsymbol{c})^2}{2\boldsymbol{\sigma}^2}\right), \tag{3}$$

where $I_{coord}$ is a tensor of image coordinates, $\boldsymbol{c} = \left(\frac{bl+br}{2}, \frac{bt+bb}{2}\right) \in \mathbb{R}^2$ is the center of the box, and standard deviation $\boldsymbol{\sigma} \in \mathbb{R}^2$ is typically set to one-third of the box height $h$ and width $w$.[3]

While effective as demonstrated in Ma et al. (2024b), the edit in Equation (2) is not differentiable with respect to the box parameters as it contains discrete borders, especially where the mask $\boldsymbol{M}_*$ transitions either from 0 to 1 or vice versa. Moreover, the shape of the edit, as shown in Figure 3, has sharp discontinuities at the edges. While the Gaussian map allows focusing the attention map to the center of the box, the standard

---

[1]Trailblazer Ma et al. (2024b) uses a per-animal hyperparameter setting, but this is impractical for our evaluation scenario with hundreds of animals. We thus use this value to a fixed value that we found empirically and use it for all evaluations, including our method.

[2]In the case of the temporal attention map, there are two centers, one for each frame, and we simply take the maximum value for each pixel for overlapping regions.

[3]The original paper claims one-half, but the official code release uses one-third, which is what we use.

deviation $\sigma$ of the Gaussian is set to a large enough value to ensure that the attention map is spread over the entire box, but this then leads to a truncated shape with clipped edges. Thus, it is non-trivial to optimize the box parameters with respect to any objective.

## 3.2 Differentiable attention map editing

To address this, we introduce a differentiable attention map editing method that does not rely on replacement. Specifically, we use the Gaussian edit map $M_{\mathcal{G}}$ as a starting point, make its borders smooth so that it is differentiable. We then propose to edit without relying on the binary masks $M_S$ and $M_T$.

**Smooth masks.** To prevent the discontinuities shown in Figure 3, we smooth out the borders of the Gaussian edit map $M_{\mathcal{G}}$, with 1D smooth step functions, both in the horizontal and vertical directions. Formally, we write

$$M_{\mathcal{B}} = M_{\mathcal{G}} \odot M_x \odot M_y, \tag{4}$$

where $M_x$ and $M_y$ are the smooth step functions defined as

$$
\begin{aligned}
M_x &= \mathrm{Sig}\left(\frac{I_u - b_l}{\kappa}\right) \odot \mathrm{Sig}\left(\frac{b_r - I_u}{\kappa}\right) \\
M_y &= \mathrm{Sig}\left(\frac{I_v - b_t}{\kappa}\right) \odot \mathrm{Sig}\left(\frac{b_b - I_v}{\kappa}\right)
\end{aligned}, \tag{5}
$$

where $\kappa$ controls the strength of the smooth edge transition, Sig denotes the standard sigmoid function, and $I_u, I_v \in H \times W$. Formally, strength $\kappa = \lambda_{edge}\sqrt{h^2 + w^2}$, is calculated as a fraction of the bounding box's diagonal length, where $h, w$ are the height and width of the box. In practice, we set $\lambda_{edge} = 0.03$. We show an example of our attention map edit in Figure 3d.

**Differentiable editing.** It is important to note that differentiability becomes limited as long as the discrete mask $M_*$ is utilized. The first term in Eq. (2) does not provide a useful gradient for optimizing the box coordinates, since the binary mask $M_*$ represents a hard selection. As a result, the derivative of $M_*$ with respect to the box coordinates is zero almost everywhere and undefined at the mask boundaries.

Although, the second term may at first glance appear differentiable, it is still evaluated under multiplication by $M_*$. Since Eq. (2) involves a product between $M_*$ and $M_{\mathcal{G}}$, applying the product rule yields one term involving the derivative of $M_*$, which vanishes almost everywhere, and another term involving derivatives of $M_{\mathcal{G}}$. While the latter term is non-zero, it only affects how values are distributed within the current box and provides limited gradient signal for moving or resizing the box itself.

Though gradients (or subgradients) may flow through the selected values, they do not provide a faithful signal for learning the selection boundaries without an explicit relaxation Xie et al. (2020). In contrast, our masks $M_{\mathcal{B}}$ are fully differentiable and of similar shape as the binary ones, and can thus be used instead of $M_*$ in Equation (2). We thus write:

$$\bar{A}_* = \underbrace{A_* \odot (\lambda_{\mathrm{w}}(1 - M_{\mathcal{B}}) + M_{\mathcal{B}})}_{\text{weakening outside}} + \underbrace{\lambda_{\mathrm{s}} M_{\mathcal{B}}}_{\text{strengthening inside}}, \tag{6}$$

where again, $\lambda_{\mathrm{w}}$ and $\lambda_{\mathrm{s}}$ are the weakening and strengthening factors, respectively, which we set empirically as $\lambda_{\mathrm{w}}=0.001$ and $\lambda_{\mathrm{s}}=0.15$ same as in the case of Trailblazer Ma et al. (2024b). While the difference between Equation (6) and Equation (2) is subtle, Equation (6) has no discrete borders, and is thus safe to differentiate with respect to the box parameters.

### 3.3 Optimizing bounding boxes to align with attention maps

With the attention map editing now being differentiable with respect to the box parameters, we can optimize the bounding box such that the attention map of the next layer is maximized within the box.[4] The difficulty in doing so, however, is how to define what is a good bounding box edit. Our hypothesis is that a good edit would be one that is 'easy' for the neural network to follow, thus being close to what the neural network itself would do. This is because both attention map edits in Section 3.2, are heuristically designed and there is no guarantee that such edit would not derail the video generation process. In fact, as shown earlier in Figure 1, this does happen.

We thus propose to look into how the edited attention maps propagate through the network layers, and optimize the bounding boxes such that this propagation follows the user's intent. Specifically, without loss of generality, let us denote the transformation induced by the next attention layer as $f$, the 'values' in the typical attention operation as $V$, we can then write

$$(A_*^{l+1}, V_*^{l+1}) = f(\bar{A}_*^l, V_*^l), \tag{7}$$

where the superscript $l$ denotes the layer index, the subscript $*$ again denotes both spatial and temporal, and $\bar{A}_*^l$ is the edited attention map of layer $l$. We then look at $A_*^{l+1}$, which now represents how the network is utilizing edited attention, and aim to encourage the attention to be within the user-specified bounding box mask $M_*$.

Inspired by cross-attention guidance (Chen et al., 2024b) , we define the loss to encourage that the sum of all attention values $\sum A_*^{l+1}$ is maximized within the box. This can be expressed in various ways, but as the magnitude of attention may differ from one layer to another, we define it such that the sum of the attention values solely come from within the box, that is $\sum A_*^{l+1} \approx \sum A_*^{l+1} \odot M_*^{l+1}$. The summation is taken over spatial positions and selected tokens; the resulting scalar is squared and averaged over the batch and across layers. We thus write our loss function as

$$\mathcal{L}_{\text{attn}} = \left\| 1 - \frac{\sum A_*^{l+1} \odot M_*^{l+1}}{\sum A_*^{l+1}} \right\|_2^2, \tag{8}$$

Note that this loss then scales between 0 and 1, with 0 indicating the attention map is purely concentrated to be within the bounding box.

**Preventing attention from completely ignoring outside the box.** Solely relying on the maximization of attention inside the bounding box causes the model to ignore the background, resulting in poor generation quality and less alignment to the prompt. Thus, we introduce a balancing loss, which allows our method to attend to the outside region, leading to an effective whole generation. We write

$$\mathcal{L}_{\neg\text{attn}} = \left\| 1 - \frac{\sum A_*^{l+1} \odot \left(1 - M_*^{l+1}\right)}{\sum A_*^{l+1}} \right\|_2^2, \tag{9}$$

which now enforces outside of the box to retain attention.

**Regularizing to remain close to user intent.** As we wish to preserve as much of the user intent as possible, we regularize to keep the box close to the original one. We write

$$\mathcal{L}_{\text{reg}} = \| b - b_{\text{user}} \|_2^2, \tag{10}$$

where $b$ would be the optimized bounding box, and $b_{\text{user}}$ the original box provided by the user.

**Final optimization objective.** The final loss is then,

$$\mathcal{L}_{\text{total}} = \mathcal{L}_{\text{attn}} + \lambda_{\neg\text{attn}}\mathcal{L}_{\neg\text{attn}} + \lambda_{\text{reg}}\mathcal{L}_{\text{reg}}, \tag{11}$$

where $\lambda_{\neg\text{attn}}$ and $\lambda_{\text{reg}}$ are the regularization strength, which we empirically set to $\lambda_{\text{reg}}{=}0.1 \times \sqrt{A}$, where $A$ is the number of pixels in the image and $\lambda_{\neg\text{attn}}{=}10$.

---

[4]Note that this alone causes the model to highly ignore outside the bounding box, which we balance; for ease in explanation, we omit this for now.

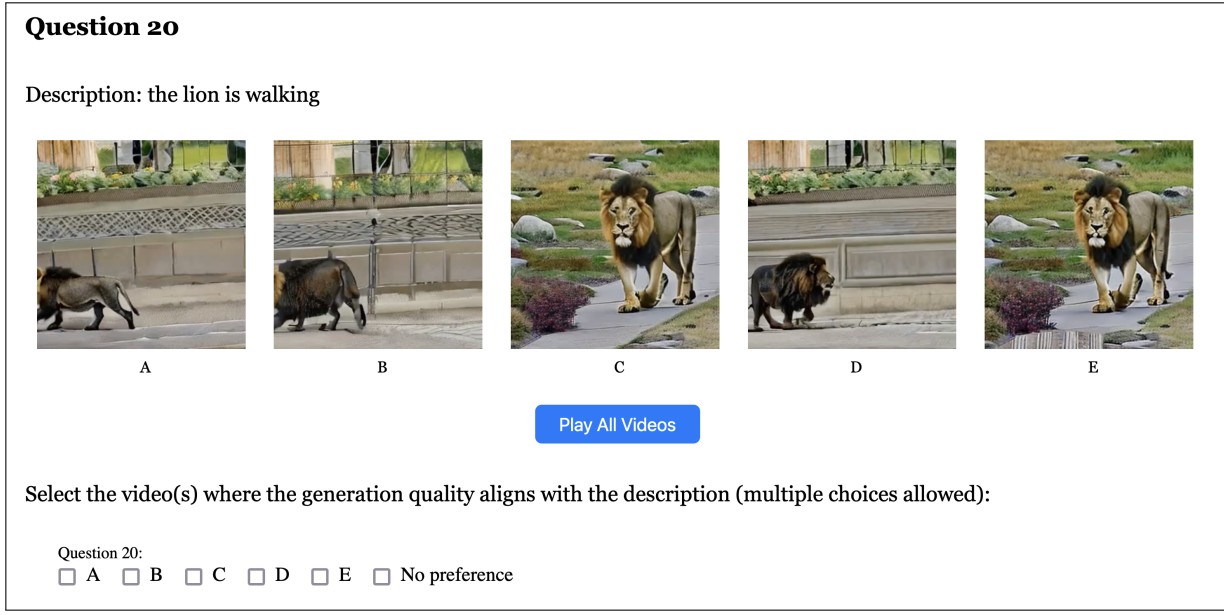

Figure 4: **User study interface** – We show a sample of the user study interface for prompt 'the lion is walking'. Users are asked to select their preference over a set of five video generations, provided in a random order. Users are allowed to select multiple choices or no preference. In this example, only the generation results are shown without user control to isolate quality vs. control. For this question, C and E look preferable, whereas the control is located at the bottom-left where the lion is at A, B, and D. C and E completely ignore user control, yet generate a preferred view of the prompt. As we are interested in controlled generation, we consider both answers to the quality preference and the trajectory faithfulness when evaluating the quality of generations.

### 3.4 Implementation details

We implement our method in PyTorch Paszke et al. (2019) based on the official code of Trailblazer Ma et al. (2024b). We further apply our method from Trailblazer Ma et al. (2024b) to T2V-Turbo Li et al. (2025), a different video generator. For a fair comparison, we use the same video length of 24 frames, and the default setting for all other hyperparameters associated with Trailblazer Ma et al. (2024b). Because of our resource constraints—memory of our GPU cards is 32GB—we use a resolution of $256 \times 320$ and $320 \times 320$ for T2V-Turbo Li et al. (2025) and Trailblazer Ma et al. (2024b), respectively. We also use $320 \times 320$ resolution for the Peekaboo baseline, and the native resolution $320 \times 512$ for Freetraj baseline. We find that $\lambda_s$=0.3 works well for the T2V-Turbo Li et al. (2025) backbone. Following Ma et al. (2024b), we only edit the spatial attention map. We run 5 optimization iterations for 5 editing steps, leading to 25 iterations in total. We use the Adam optimizer Kingma & Ba (2015) for box adjustments and provide further details on our optimization procedure in the supplemental material.

## 4 Experiments

### 4.1 Experimental setup

Evaluating controllable T2V models is challenging as generated videos lack ground truth and quality metrics only form a proxy to human-perceived quality. It is also non-trivial to generate large and diverse, input control trajectories without violating physics, as the generation outcome is highly sensitive to the input control, as we will demonstrate. For the former, we run a user study alongside human-preference metrics. For the latter, we use control trajectories from animal videos. In the following subsections, we will first

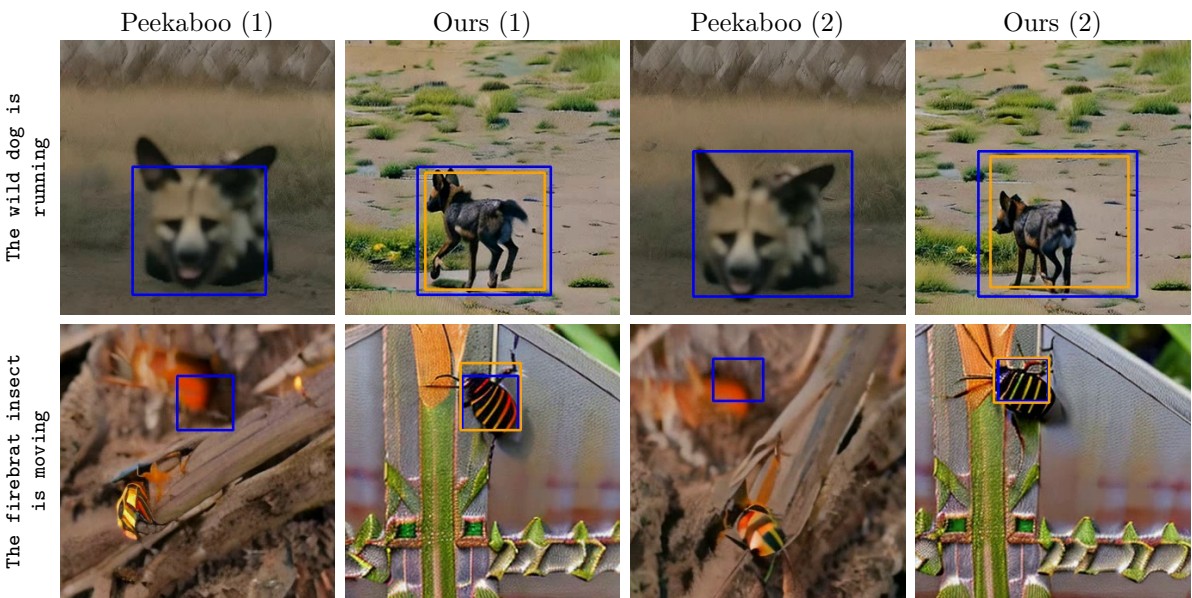

Figure 5: **Qualitative results comparing Peekaboo Jain et al. (2024) and our method (full).** –
Each row shows two representative frames per method (left to right: Peekaboo, Ours). Our method yields better
generation quality and improved alignment with the text prompt.

discuss the experimental setup, including the dataset and how we create the control trajectories. We also
discuss the baselines, the quantitative metrics, and the user study protocol.

**Dataset and bounding-box trajectories.** As in Ma et al. (2024b), we use the Animal Kingdom dataset Ng
et al. (2022) as a reference dataset for actual videos. This dataset contains several wild animals such as
cheetah and hippotamus, with corresponding text describing activity, e.g., *the cheetah is running.* This
dataset contains 18,744 video clips.

We keep only video clips with a single object using NLTK library Bird et al. (2009), and moving verbs
indicating motion. We then use OWL large-scale open vocabulary detector Minderer et al. (2022) to detect
object boxes, PySceneDetect Castellano (2014–2024) library to split scenes to extract trajectories, i.e., se-
quence of bounding boxes. To obtain trajectories that are useful, we drop videos that have less than two
frames with the object being detected, or with only two detections across scene cuts. Also, videos that show
discontinuous trajectories such as abrupt camera motions or frame cuts are dropped. These are trajectories
that have Intersection over Union (IoU) between the consecutive frames being less than 50. The above
results in 1,980 videos (157 unique animals) for training and 526 videos (96 unique animals) for testing. We
interpolate detected boxes for the missing frames to obtain a complete trajectory over the video. We further
filter out trajectories that are of boxes that are too small, i.e., maximum box width and height smaller than
10% of the width and height of the image, and then randomly sample an interval of 24 frames from the
trajectory. This results in 377 trajectories from the training set that can be used for method design and
validation, and 226 trajectories exclusively for testing.

**Baselines.** We compare our method against Peekaboo Jain et al. (2024), showcasing the quality improve-
ment of our method. We also compare against Trailblazer Ma et al. (2024b), with its original backbone and
we further adapt it to a different recent text-to-video model T2V-Turbo Li et al. (2025). We also compare
our method against different variations of our method, specifically, our method without box optimization,
our method without Equation (9), and using our optimized boxes with Trailblazer Ma et al. (2024b).

**Quantitative metrics.** We quantify the performance of each method using human-preference alignment
metrics. We report PickScore Kirstain et al. (2023) and HPS v2 (Human Preference Score) Wu et al. (2023).
We also report the mean intersection over union (mIOU) between the detected objects via an open-vocabulary
detector Minderer et al. (2022) and the user control, using the animal kingdom dataset as reference.

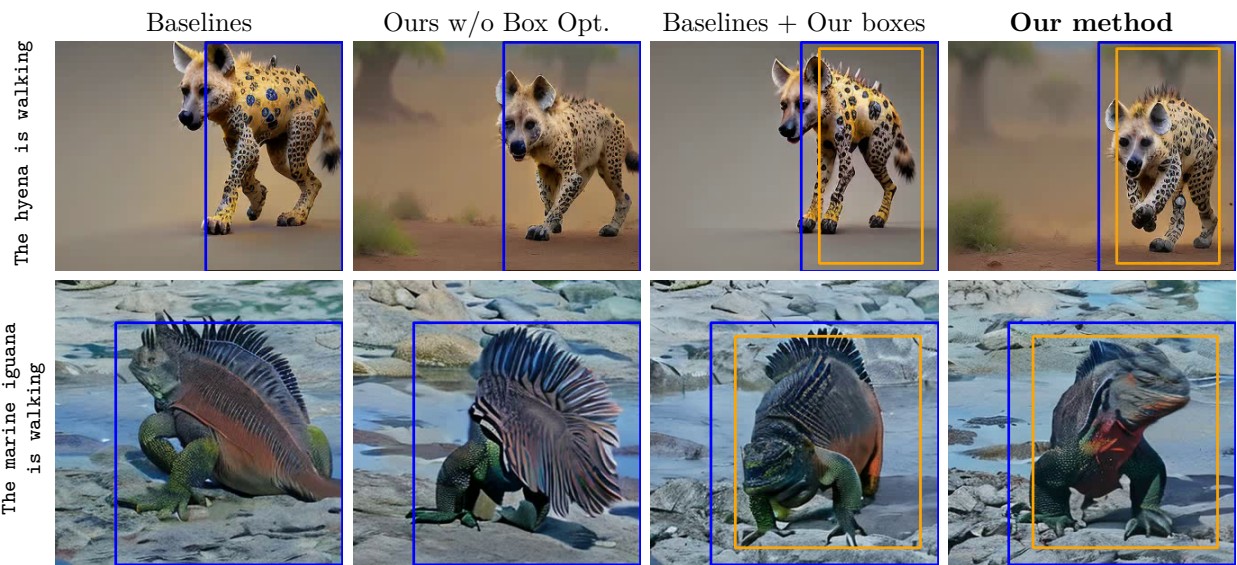

Figure 6: **Qualitative results with T2V-Turbo Li et al. (2025) (top row) and Trailblazer Ma et al. (2024b) (bottom row) backbones.** – We display the original user control in blue and our optimized boxes in orange. As shown, ours provides improved rendering quality and adherence to control and the prompt, shown in the top and bottom row. See video results for best viewing.

**User study evaluation**. We also conduct a user study with 20 participants selecting their preference over a set of 5 different methods; see Figure 4. Users were asked to perform two different tasks: (1) to evaluate the quality of the generations without being provided any bounding-box annotations to solely evaluate quality; and (2) to evaluate the adherence to the bounding box. These were provided as two separate questions and the questions were grouped by tasks. Multiple choices were allowed, including no preference. Each participant was asked to answer 40 questions (20 for T2V-Turbo Li et al. (2025) and 20 for Trailblazer Ma et al. (2024b)) which takes about 20–30 minutes.

To rule out generations that completely failed due to the limitations of the video diffusion model, we chose only those which provide an IoU value of at least 60% for *at least one of the methods* being compared. Candidate questions were then selected randomly and anonymized. A final set with overall quality is used for the study. This random selection was fixed for all participants.

Finally, as the first task ignores the user control completely (following user study design), the preference recorded for this tasks can be irrelevant to the controlled generation task as shown in Figure 4. Thus, we mark as preferred quality only when user selected the video for both tasks. This study was approved by the Institutional Review Board.

## 4.2 Results

**Quantitative results.** We report the standard quantitative results in Table 1. Overall, our method demonstrates consistent improvements across different architectures Ma et al. (2024b); Li et al. (2025) (top and bottom row). Also, our method outperforms baselines such as Peekaboo Jain et al. (2024), Trailblazer Ma et al. (2024b), and Freetraj Qiu et al. (2024) in terms of human preference scores using PickScore and HPSv2 (bottom row). Additionally, *"Our boxes + Trailblazer Ma et al. (2024b) "* validates the benefits of applying our adjusted boxes (top row). In terms of control accuracy (mIOU), the results show that our method matches the same performance as the baselines. Though the benefit of adjustments is well highlighted in the user studies, this results reveals that our adjustments do not degrade control.

**Qualitative results.**

| Model | PickScore ↑ | HPSv2 ↑ | mIOU ↑ |
|---|---|---|---|
| Trailblazer Ma et al. (2024b) | 0.244 | 0.222 | 0.37 |
| **Our boxes** + Trailblazer backbone | 0.257 | 0.223 | 0.36 |
| **Our method** w/o Box Opt. | 0.243 | 0.221 | 0.37 |
| **Our method** (full) | 0.257 | 0.225 | 0.37 |
| Peekaboo Jain et al. (2024) | 0.125 | 0.189 | 0.30 |
| Trailblazer Ma et al. (2024b) | 0.146 | 0.222 | 0.37 |
| Freetraj Qiu et al. (2024) | 0.178 | 0.223 | 0.34 |
| Trailblazer + T2V-Turbo backbone | 0.234 | 0.253 | 0.41 |
| **Our method** using T2V-Turbo backbone | 0.317 | 0.263 | 0.41 |

Table 1: **Quantitative results with human-preference and control metrics** – Our full method outperforms baseline and demonstrates consistent preference across different architectures, while achieving competitive control.

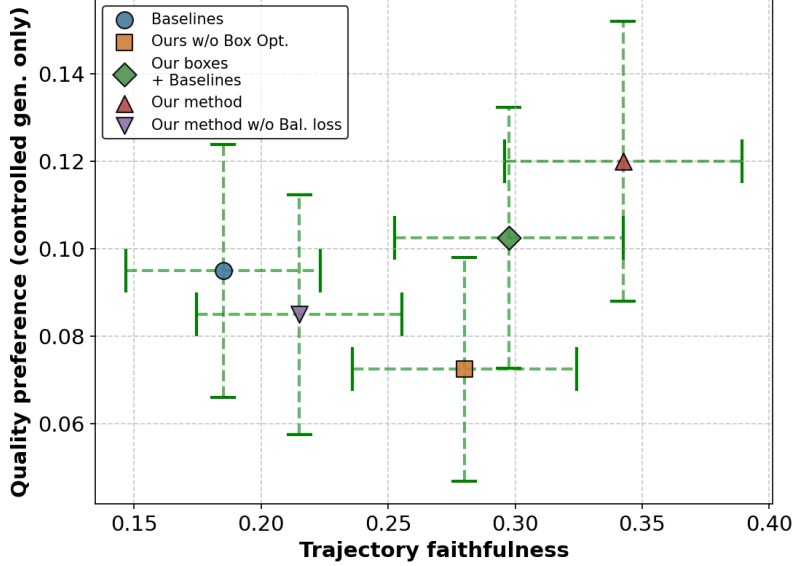

Figure 7: **User study results** – We report the user preference for trajectory faithfulness on the x-axis, and quality of generation (for generations that adhere to control) on the y-axis. The 95% confidence intervals are drawn as green dashed lines. Our method significantly outperforms Trailblazer Ma et al. (2024b) with both backbones.

We show qualitative examples of video frames generated by our method compared to baselines in Figure 5, Figure 6, and Figure 8. Notably, our method performs significantly better than Peekaboo Jain et al. (2024) and Freetraj Qiu et al. (2024), in terms of generation quality and adherence to control, in Figure 5 and Figure 8 respectively.

Also, when compared with T2V-Turbo and Trailblazer backbones, our method, courtesy of adjustments, produces better generation that adhere with the prompt in Figure 6. This is especially visible in the second row, where generation follows motion in the prompt *"The marine iguana is walking"*. Interestingly, applying our adjusted bounding boxes also provides better prompt following for the baseline method. Though this benefit is not always transferable, as the baseline still uses a discrete editing strategy. Best performance

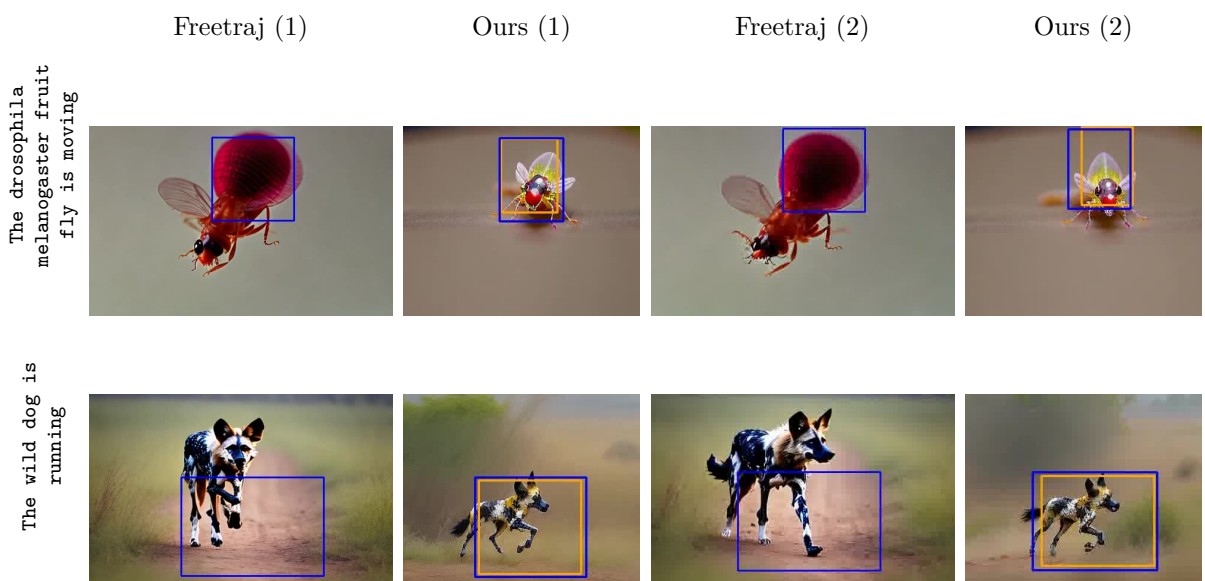

Figure 8: **Qualitative results comparing Freetraj Qiu et al. (2024) and our method on T2V-Turbo Li et al. (2025).** Frames are shown at a common display height (aspect ratio preserved); frame resolutions differ (Freetraj: 320×512, Ours: 256×320). Ours still delivers better generation quality and adherence to control.

with adjustment is achieved with our method. Finally, in Figure 9 and Figure 10, we show additional qualitative results of our method compared to baselines.

**User evaluation.** We report user study results in Figure 7. As shown, our method is the most preferred by a large margin. It is also worth noting that using our adjusted boxes also benefits Trailblazer Ma et al. (2024b), as shown by the slight increase in preference. We hypothesize that this is because our boxes are optimized for the same underlying model. This further demonstrates that small adjustments matter, not only for the method that we have introduced, but also for the base model. Finally, Ours w/o balancing loss (i.e., Equation (9) being left out) show that taking the overall generation into account is effective. Simply optimizing the bounding box, without considering the whole generation results in a middle ground between the baseline and the non-optimized version. However, when both are enabled, we can achieve a significant performance increase.

## 5   Conclusion

In this work, we demonstrated that small adjustments to user-provided bounding boxes can lead to significant improvements in controlled video generation. By optimizing the bounding boxes to better align with the internal attention maps of video diffusion models while maintaining proximity to user inputs, we achieved both higher quality generations and better adherence to control signals. Through extensive experiments and user studies, we validated that our simple yet effective approach outperforms existing methods for controlled video generation. We believe our findings open up new possibilities for improving user control in text-to-video generation by considering how control signals can be optimized to work better with pretrained models.

**Limitations and future work** A limitation of our method is that the quality of generated images can be bound by the underlying video model, as shown in Figure 11. With the rapid progress in this area, we believe this limitation will be alleviated naturally. While in theory our method can be applied to other forms of control, we have only validated our idea to bounding boxes. But beyond bounding boxes, our work highlights a broader principle: control signals themselves are often misaligned with a model's internal representations, and modest optimization can substantially improve outcomes. This perspective naturally extends to future work on optimizing other conditional inputs, such as trajectories, sketches, or depth cues. Finally, our method requires partial back-propagation during optimization, thus slows down generation. With the T2V-Turbo Li

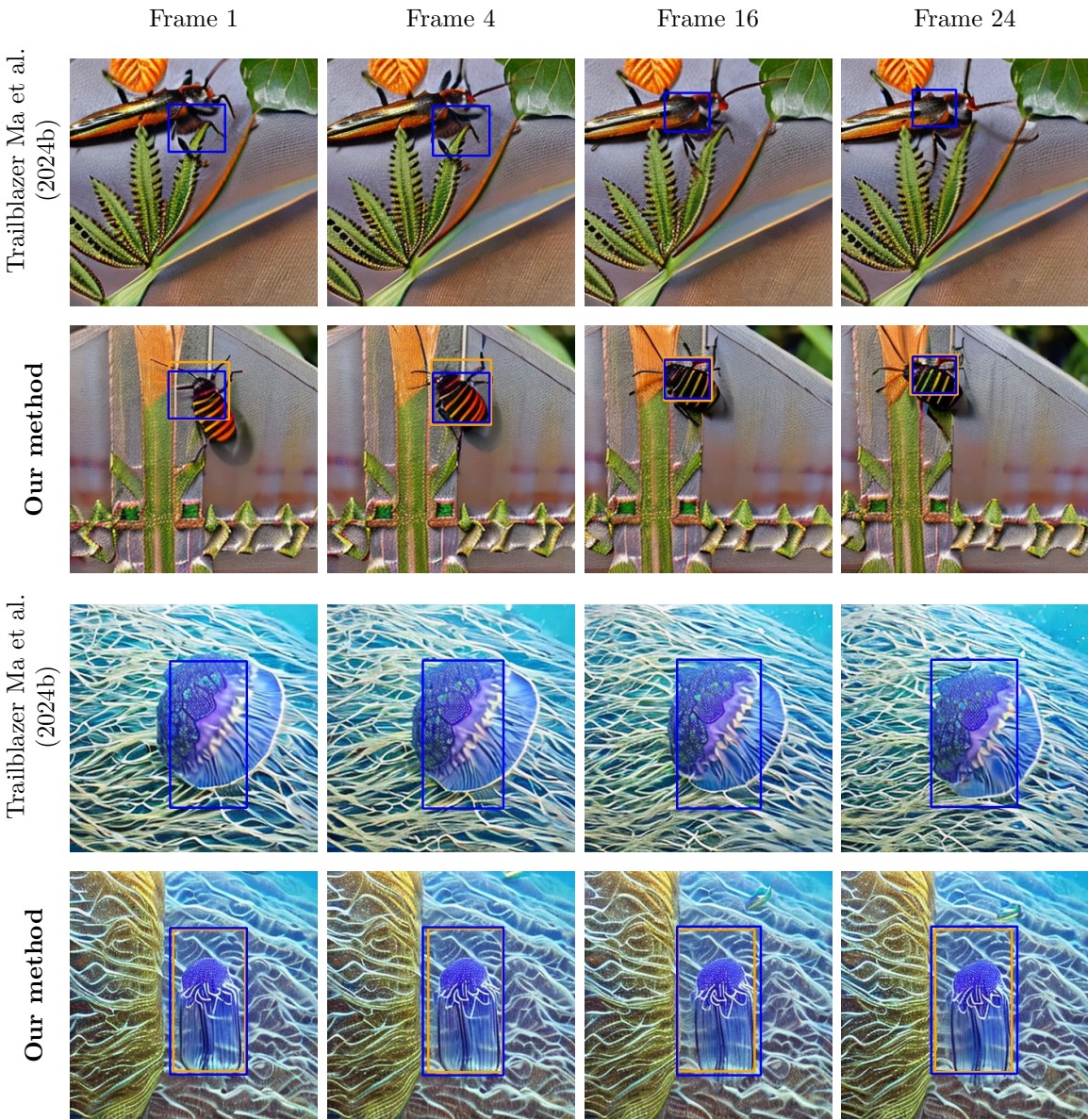

Figure 9: **Additional qualitative results with Trailblazer Ma et al. (2024b)**– We show additional examples of our edits. As shown, ours is more consistent with user intent and of higher quality. The prompt for the top 2 rows is *"The firebrat insect is moving"*, while the prompt for the bottom 2 rows is *"The jellyfish is swimming"*. Video results are available in the supplementary local HTML page.

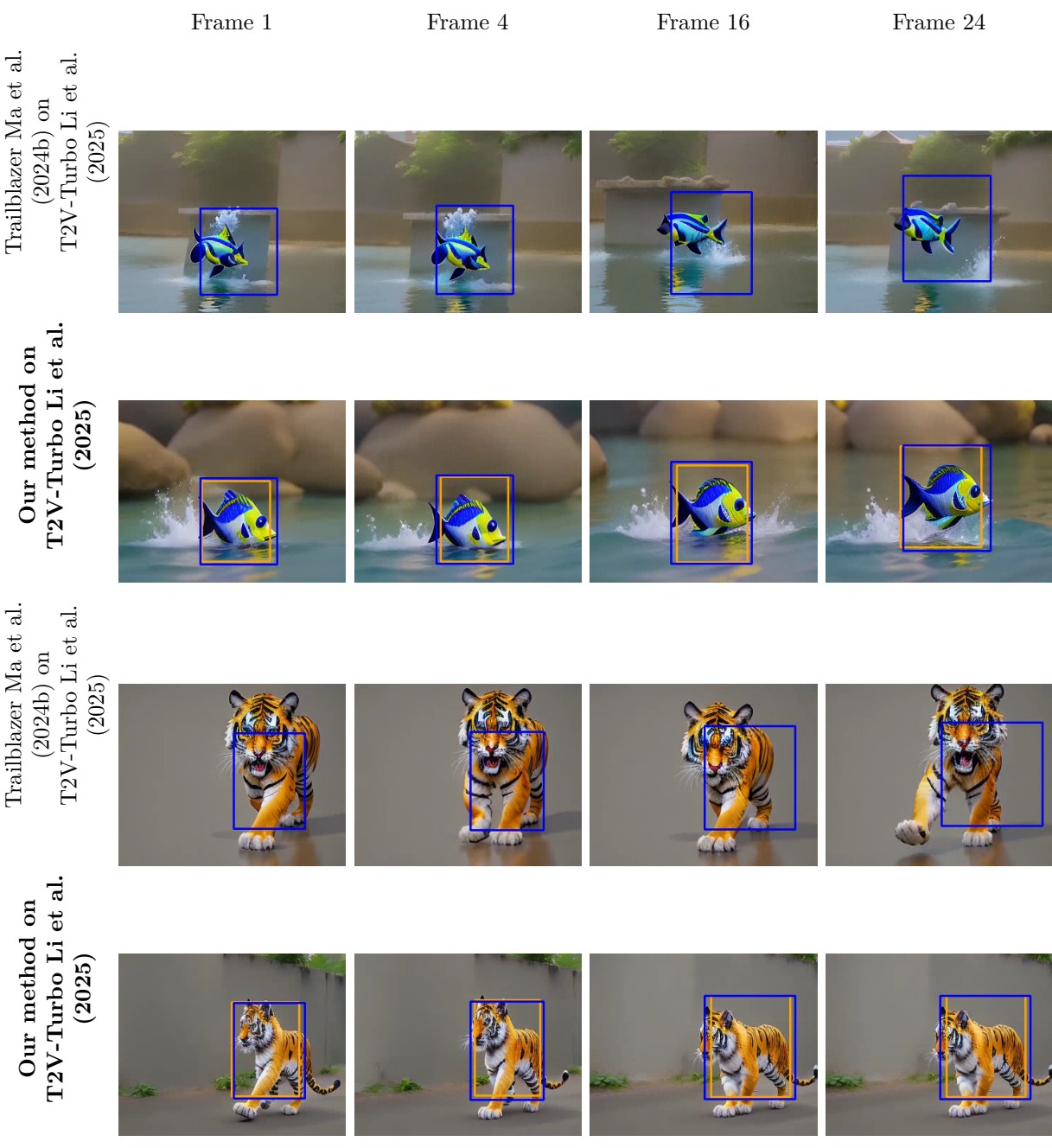

Figure 10: **Additional qualitative results with T2V-Turbo Li et al. (2025) backbone** − We show additional examples of our edits. Ours show more consistency with user intent and is of higher quality within the box. The prompt for the top 2 rows is *"The surgeonfish is swimming"*, while the prompt for the bottom 2 rows is *"The tiger is walking"*. Video results are also available in the supplementary local HTML page.

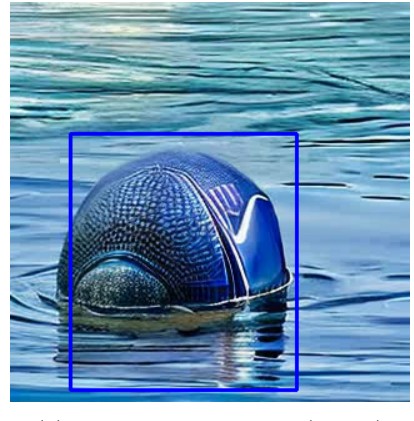 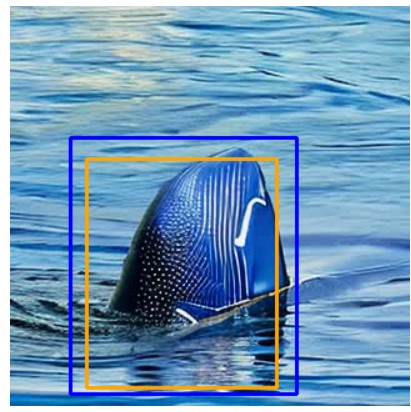

(a) Trailblazer Ma et al. (2024b)          (b) Our method

Figure 11: **Failure cases** − We show example failure cases when the video model fails to generate the desired output for the prompt `the orca is swimming`. Here, regardless of the bounding box input, the model is unable to generate the content as it has no knowledge of the target prompt. We hypothesize that more advanced video models, which are being made increasingly available Research (2023); Wan et al. (2025), will alleviate this problem.

et al. (2025) backbone, our generations take one minute and 37 seconds, while Trailblazer + original T2V Turbo takes 48 seconds, both on an NVIDIA RTX A6000. While our edits bring significant enhancements, optimizing this would also be an interesting extension. Conclusively, our main novelty demonstrates that small adjustments matter, and we therefore wish to draw the attention of the community to the sensitivity of control signals and hope that our work will serve as a foundational ground.

**Broader impact statement** Our method is applicable beyond animals, and generations can be abused with malicious intent. For example, controlling the movement of celebrities for personal gain. We discourage such practices and hope that providing proper awareness can alleviate or minimize variations of such intent.

**Acknowledgement** This work was supported by Borealis AI through the Borealis AI Global Fellowship Award. We acknowledge the computational resources provided by the University of British Columbia and thank the volunteers who participated in the user study.

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

# Supplementary Material

We provide additional information to the main content, including an ablation study, failure cases, further implementation details, and results with complex patterns and difficult trajectories. We also include more results, in terms of videos, as well as an interactive slider that visualizes how internal attention maps evolve during optimization, on the project webpage.

## A   Ablation study

**Regularization strength for deviation.** We ablate the impact of the regularizer strength in Equation (11) for how much the box deviates from user intent. We show our results with Trailblazer Ma et al. (2024b) in Figure 12. As shown, stronger regularization leads to bounding boxes that are more faithful to the user input, but their generation results are poor in quality. Too low values create too much deviation. $\lambda_{\mathrm{reg}}{=}0.1$ strikes a good balance between user intent and generation quality.

**Linear trajectories.** We also show our method applied to linear trajectories to simulate a user input. In Figure 13, we use T2V-Turbo Li et al. (2025) with our method and provide a bounding-box that linearly interpolates between a start and an end bounding box. As shown, our method provides enhancement.

| Attention map | Attention reshaped | # Edit channels | Edited copy (lower channels) | Layer info |
|---|---|---|---|---|
| 240, 1600, 77 | 240, 40, 40, 77 | 120 | 120, 40, 40, 77 | down_blocks.0.0.0 |
| 240, 1600, 77 | 240, 40, 40, 77 | 120 | 120, 40, 40, 77 | down_blocks.0.1.0 |
| 480, 400, 77 | 480, 20, 20, 77 | 240 | 240, 20, 20, 77 | down_blocks.1.0.0 |
| 480, 400, 77 | 480, 20, 20, 77 | 240 | 240, 20, 20, 77 | down_blocks.1.1.0 |
| 960, 100, 77 | 960, 10, 10, 77 | 480 | 480, 10, 10, 77 | down_blocks.2.0.0 |
| 960, 100, 77 | 960, 10, 10, 77 | 480 | 480, 10, 10, 77 | down_blocks.2.1.0 |
| 960, 25, 77 | 960, 5, 5, 77 | 480 | 480, 5, 5, 77 | mid_block.0.0 |
| 960, 100, 77 | 960, 10, 10, 77 | 480 | 480, 10, 10, 77 | up_blocks.1.0.0 |
| 960, 100, 77 | 960, 10, 10, 77 | 480 | 480, 10, 10, 77 | up_blocks.1.1.0 |
| 960, 100, 77 | 960, 10, 10, 77 | 480 | 480, 10, 10, 77 | up_blocks.1.2.0 |
| 480, 400, 77 | 480, 20, 20, 77 | 240 | 240, 20, 20, 77 | up_blocks.2.0.0 |
| 480, 400, 77 | 480, 20, 20, 77 | 240 | 240, 20, 20, 77 | up_blocks.2.1.0 |
| 480, 400, 77 | 480, 20, 20, 77 | 240 | 240, 20, 20, 77 | up_blocks.2.2.0 |
| 240, 1600, 77 | 240, 40, 40, 77 | 120 | 120, 40, 40, 77 | up_blocks.3.0.0 |
| 240, 1600, 77 | 240, 40, 40, 77 | 120 | 120, 40, 40, 77 | up_blocks.3.1.0 |
| 240, 1600, 77 | 240, 40, 40, 77 | 120 | 120, 40, 40, 77 | up_blocks.3.2.0 |

Table 2: **Edited cross attention maps** – As in Trailblazer Ma et al. (2024b), we edit only the lower channels across all layers. Each layer follows the spatial cross-attention naming where *down_blocks.0.0.0* can be fully written as ***down_blocks.0**.attentions.**0**.transformer_blocks.**0**.attn2*.

## B   Failure cases

Additionally, 11% and 33% of the users did not have a preference for the quality and trajectory faithfulness questions respectively. This can be attributed to the limitations of the underlying video model both in terms of inability to faithfully generate according to the prompt and at desired locations. Still, with more advanced video models becoming increasingly available Research (2023); Wan et al. (2025), we expect this problem to be mitigated in the future.

## C  Further implementation details

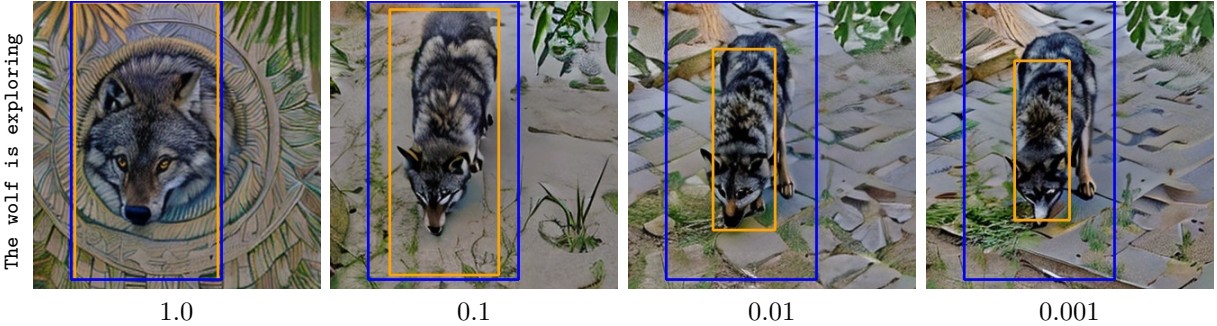

Figure 12: **Effect of regularization strength/penalty** − We show an example frame with original bounding boxes as blue and the adjusted boxes as orange. A higher penalty (1.0) allows less movement, while too loose penalty (0.001) causes generation to stray away from user intent. We set the penalty to 0.1 for all our experiments, which strikes a good balance.

**Optimization procedure** We present our optimization procedure in Algorithm 1 and show how optimization interleaves with the diffusion process.

---

**Algorithm 1** Trajectory optimization

---

1: **Input:** Video diffusion model $\Theta_b$, decoder $D$, initial latents $z$, prompt embedding $p$, denoise steps $K$, spatial edit steps $K_S$, temporal edit steps $K_T$, inner optimization steps $O$, learning rate $\sqcap$, and trajectory boxes $\mathcal{B}$
2: **for** $i \leftarrow 1$ to $K$ **do**
3:     $r \leftarrow 0$             ▷ inner optimization step counter
4:     $t \leftarrow \text{TimestepSchedule}(i)$
5:     **while** $(i < K_S$ or $i < K_T)$ and $r < O$ **do**
6:         /* extract attention maps and noise prediction */
7:         $(A_S, A_T, \epsilon_t) \leftarrow \Theta_b(p, z, t)$
8:         /* Apply editing (Eq. 6), compute loss (Eq. 11) and gradients */
9:         $(\tilde{A}_S, \tilde{A}_T) \leftarrow \text{edit}(A_S, A_T)$     ▷ Eq. 6
10:        $\mathcal{L} \leftarrow \mathcal{L}(\tilde{A}_S, \tilde{A}_T, \mathcal{B})$     ▷ aggregated over frames and layers; Eq. 11
11:        $g \leftarrow \nabla_{\mathcal{B}} \mathcal{L}$
12:        /* Update bounding boxes */
13:        $\mathcal{B} \leftarrow \text{AdamStep}(\mathcal{B}, g; \sqcap)$
14:        /* Project boxes to valid boundaries */
15:        $\mathcal{B} \leftarrow \text{Proj}(\mathcal{B})$
16:        $r \leftarrow r + 1$
17:     **end while**
18:     /* standard denoise step (no editing) */
19:     $\epsilon_t \leftarrow \Theta_b(p, z, t)$
20:     $z \leftarrow \text{SchedulerStep}(z, \epsilon_t, t)$
21: **end for**
22: **return** $D(z)$

---

**Keeping bounding boxes within images.** To prevent optimization from making bounding boxes go out of images, we simply clip the optimized bounding box coordinates.

**Additional hyperparameters.** Besides the hyperparameters noted in the main paper, for others, we follow Trailblazer Ma et al. (2024b) to be comparable. Specifically for Trailblazer Ma et al. (2024b), we use

Zeroscope v2 576w as the underlying text-to-video model,[5] and the same DPM Solver[6] for both methods, with a guidance scale of 9.0. We run 40 denoising steps, applying the editing (and our optimization) on the first five steps. For Peekaboo baseline, we run the same number of denoising steps with the default number of frozen steps set to 2. For Freetraj baseline, we run with 50 denoising steps. We also use their DDIM eta of 0, with 6 edit steps, and set the video length to 24 frames, matching the input trajectory.

For T2V-Turbo Li et al. (2025), we use the official implementation, with the default parameters and 16 denoising steps. We apply the edits, both of our method and Trailblazer Ma et al. (2024b) for only the first five steps.

**Control metric.** For computing the mIoU, we simply choose, among the detected bounding boxes, one that matches most closely to the user control bounding box. Note that this is different from Trailblazer Ma et al. (2024b) which relies on the object detection score. Our strategy takes into account that generated videos are not constrained to generate only a single object, as there may be distractors. We found that Trailblazer Ma et al. (2024b) strategy of using the detection scores does get easily distracted by distractors, thus providing an unfair comparison. Our strategy fixes this.

**Model architecture.** For Trailblazer Ma et al. (2024b), we use the U-Net architecture of Zeroscope. An initial gaussian noise $z_t$ with shape $1 \times 4 \times F \times 40 \times 40$ is generated where $F = 24$ frames. We edit only the spatial attention with a target video resolution of $320 \times 320$. The corresponding attention maps are computed from the queries and keys, while the total number of tokens is 77 Radford et al. (2021); each token has a dimension of 1024. The progression of the edited attention maps with their respective dimensions is detailed in Table 2.

For T2V-Turbo Li et al. (2025), the base model utilizes the same U-Net-style architecture as Trailblazer Ma et al. (2024b), hence we edit in the same way as for Trailblazer Ma et al. (2024b). We use PyTorch Paszke et al. (2019) gradient checkpointing in our implementations.

# D   Ablations on component design choices

We include comprehensive ablations on key design choices in Tables 3 to 6.

**Deviation penalty.** Table 3 shows that excessive deviation leads to loss of user intent (lower mIoU), while overly restrictive penalties limit quality improvements. A penalty of 0.1 balances control and generation quality.

**Smoothing kernel normalization.** Table 4 demonstrates that normalizing the smoothing kernel to produce consistent peak attention across layers significantly improves PickScore, HPSv2, and mIoU.

**Background preservation term.** Table 5 confirms that the background attention term is necessary for maintaining scene coherence.

**Edge strength.** Table 6 shows that very low edge strength behaves similarly to discontinuous masks (e.g., Trailblazer), degrading quality, while overly strong edges reduce control accuracy. Intermediate values (0.001–0.03) provide the best trade-off.

# E   Complex patterns and difficult trajectories

We show results for complex patterns and difficult trajectories such as morphing, zig-zag, U-turn, and stationary-to-move in Figures 14 to 16.

---

[5]Zeroscope v2 576w model. `https://huggingface.co/cerspense/zeroscope_v2_576w`

[6]Cheng Lu *et.al.*, DPM-Solver: A Fast ODE Solver for Diffusion Probabilistic Model Sampling in Around 10 Steps, *Advances in Neural Information Processing Systems*, 2022

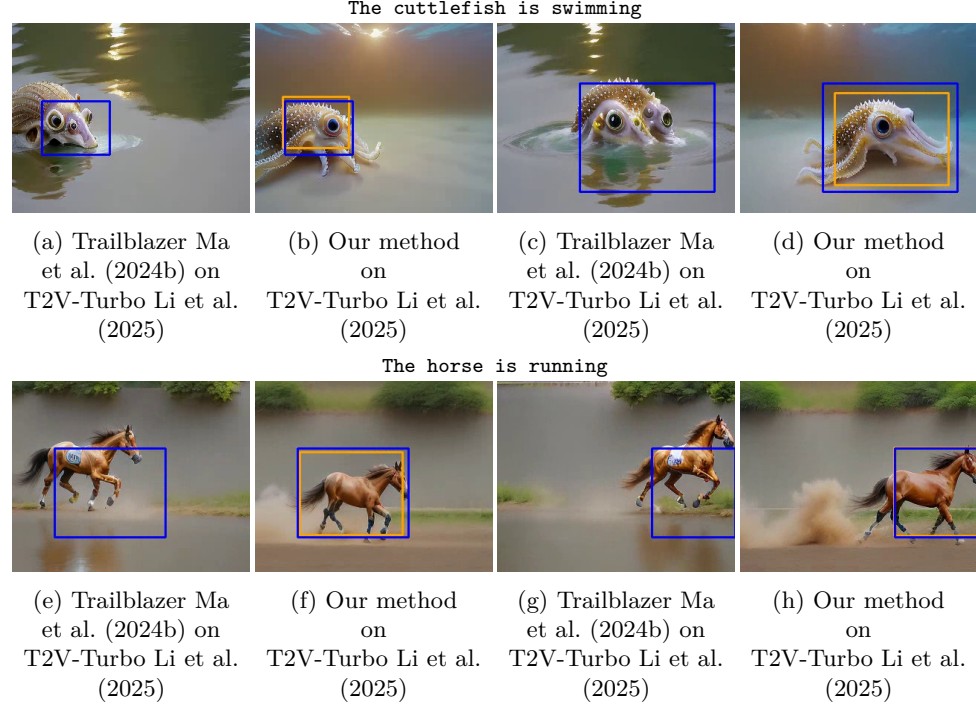

Figure 13: **Application to perspective and linear trajectories (top and bottom respectively)** – We show example frames of applying our method to T2V-Turbo Li et al. (2025) with a trajectory that linearly interpolates two bounding boxes (start and end). Ours still delivers improvements, both in terms of generation quality within the bounding box and adherence.

| $\alpha$ | PickScore ↑ | HPSv2 ↑ | mIOU ↑ |
|---|---|---|---|
| 0 | 0.210 | 0.230 | 0.31 |
| 0.001 | 0.211 | 0.230 | 0.32 |
| 0.01 | 0.199 | 0.228 | 0.36 |
| 0.1 | 0.194 | 0.225 | 0.38 |
| 1 | 0.187 | 0.221 | 0.37 |

Table 3: Ablation results for deviation penalty term $\lambda_{\mathrm{reg}} = \alpha \times \sqrt{A}$

| Normalization | PickScore ↑ | HPSv2 ↑ | mIOU ↑ |
|---|---|---|---|
| True | 0.509 | 0.225 | 0.38 |
| False | 0.491 | 0.223 | 0.38 |

Table 4: Ablation results for smoothing kernel normalization

| $\lambda_{\neg\mathrm{attn}}$ | PickScore ↑ | HPSv2 ↑ | mIOU ↑ |
|---|---|---|---|
| 0 | 0.228 | 0.219 | 0.36 |
| 1 | 0.231 | 0.221 | 0.39 |
| 10 | 0.256 | 0.225 | 0.38 |
| 100 | 0.285 | 0.228 | 0.31 |

Table 5: Ablation results for background preservation term $\lambda_{\neg\mathrm{attn}}$

| $\lambda_{edge}$ | PickScore ↑ | HPSv2 ↑ | mIOU ↑ |
|---|---|---|---|
| 0.0001 | 0.240 | 0.224 | 0.40 |
| 0.001 | 0.248 | 0.226 | 0.38 |
| 0.03 | 0.243 | 0.225 | 0.38 |
| 0.1 | 0.268 | 0.232 | 0.36 |

Table 6: Ablation results for edge strength $\lambda_{edge}$

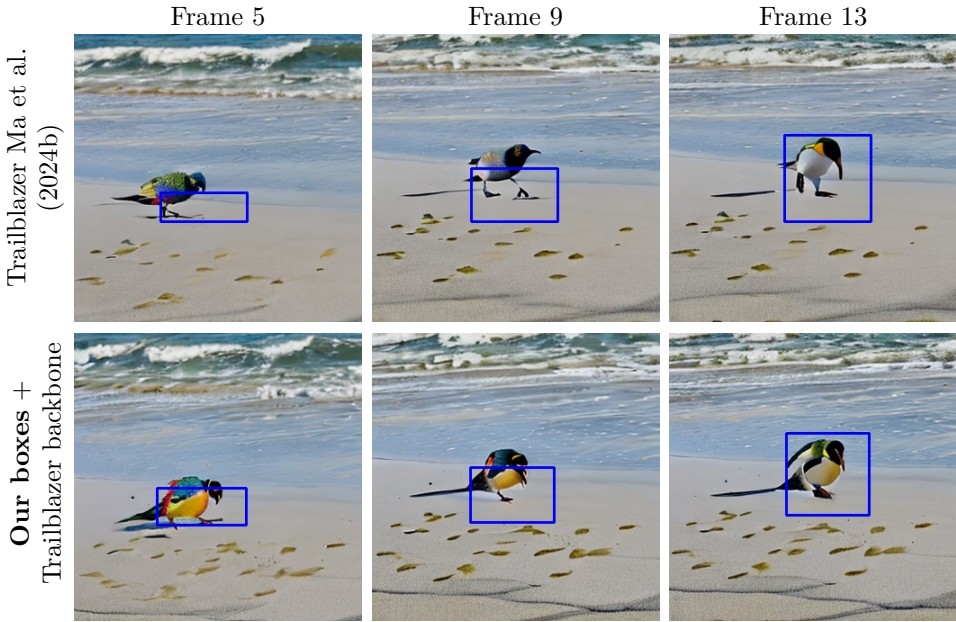

Figure 14: **Additional qualitative results for morphing task (Parrot → Penguin)−** We show transition frames, where our optimized boxes provide benefit for better control of the morphing task.

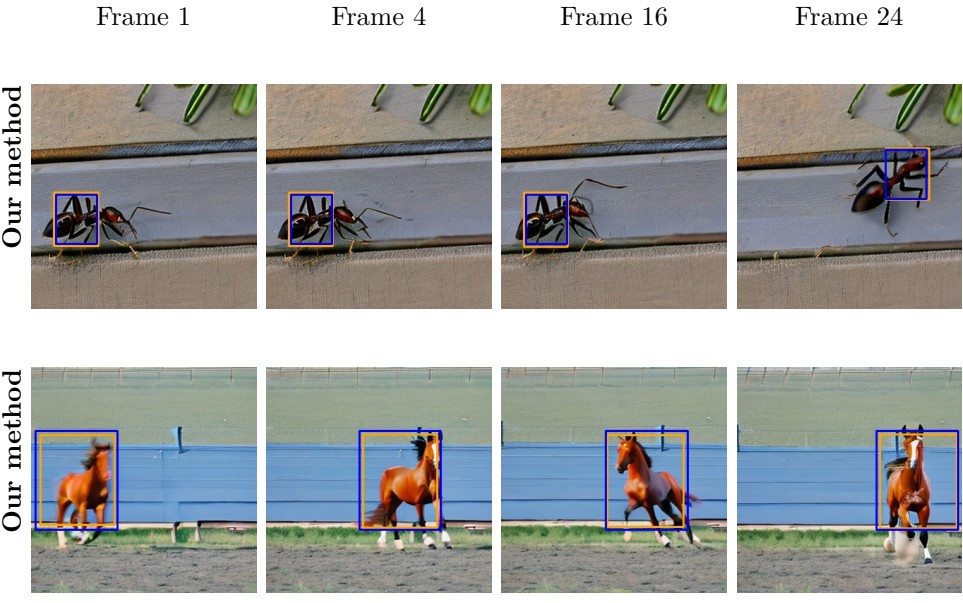

Figure 15: **Additional qualitative results with stationary-to-move and U-turn trajectories −** We show example frames where our method is applied to complex patterns and preserves difficult motions. Best viewed in motion on the project webpage.

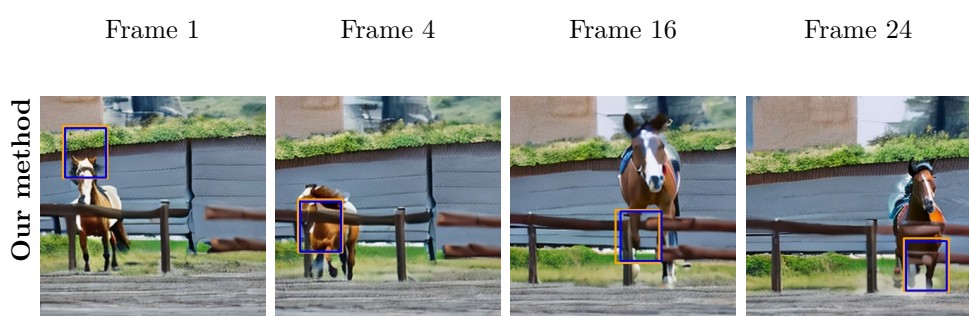

Figure 16: **Extreme scenario case with very small boxes and in a zig-zag trajectory** – We present an extreme scenario with zig-zag motion using very small boxes, different from the object size. Our method compensates and generates motion that follows user intent. Best viewed in motion on the project webpage.

