# OpenReview forum: "Making Video Models Adhere to User Intent with Minor Adjustments"
_TMLR — Accepted by TMLR_

### Review · Reviewer_tqFk · 2025-12-12

**Summary Of Contributions:**

### **Summary**
This paper proposes a neat and practical idea: instead of strictly enforcing user-provided bounding boxes for text-to-video (T2V) models, slightly *optimizing* these boxes can significantly improve both generation quality and control adherence. The key challenge is that prior box-based attention editing methods are non-differentiable with respect to box parameters. To address this, the authors introduce a differentiable attention edi and formulate an optimization objective that adjusts bounding boxes to better align with the model’s internal cross-attention behavior. Importantly, the method stays close to the original user intent while yielding noticeable gains. Experiments on the Animal Kingdom dataset, including user studies, show consistent improvements over prior work.

### **Strengths**
* The motivation is clear and intuitive: small mismatches between user control and the model’s internal "comfort zone" can hurt generation quality, and adjusting the control itself is a reasonable fix.
* While optimization-based ideas are not new (e.g., latent optimization), applying them directly to *conditional inputs* like bounding boxes is a fresh and interesting direction, especially for video.
* The proposed differentiable attention editing is simple yet effective, and integrates well with existing training-free control methods.
* Qualitative results and user studies convincingly demonstrate improved visual quality and better prompt adherence compared to baselines.

### **Weaknesses**
* Although the paper compares against strong baselines (e.g., Trailblazer), several closely related works mentioned in the related work section, such as Motion-Zero or Ctrl-V, are not experimentally discussed. It would help to clarify why these methods are not directly comparable, or to provide a more explicit discussion of how the proposed approach differs from or improves upon them.
* The claim that Equation (2) is non-differentiable with respect to the box parameters could be explained more clearly. While the discreteness can introduce discontinuities, this is not immediately obvious from the equations alone, since the box coordinates also appear in the Gaussian term. A more explicit explanation of where and why gradients break (e.g., due to hard mask boundaries) would improve clarity.
* Some notations and implementation details are underspecified. For example, the meaning of $\text{Sig}$ in Equation (5) is not explicitly defined. Additionally, the optimization setup is somewhat vague: Equation (11) depends on the layer index $l$ and diffusion timestep $t$ (although not annotated), but it is unclear which layers are optimized, over which denoising steps, and whether this is applied uniformly across the network. Clarifying these points would make the method easier to reproduce and better understood by readers.

**Audience:**

Yes

**Audience Explanation:**

Controllable video generation is a pretty active topic right now, both in research and in real applications. This paper’s main angle, improving controllability and quality by *optimizing the control signal itself*, feels like a simple but intriguing idea. I think at least part of the TMLR audience (especially folks working on diffusion-based generative modeling, controllability, and training-free guidance/editing) would be interested in these findings, and it could also spark follow-up work on optimizing other kinds of conditional inputs beyond boxes.

**Broader Impact Concerns:**

No explicit broader impact or ethical concerns raised by this work.

**Claims And Evidence:**

No

**Claims Explanation:**

Overall, the paper provides solid empirical evidence for most of its claims, but not all of them are equally convincing.

The authors list five main contributions:

1. Small adjustments to user-provided bounding boxes can significantly improve generation quality.
2. These adjustments remain close to the original user intent.
3. Existing attention-based box editing formulations are non-differentiable.
4. A new differentiable attention editing formulation is proposed to enable backpropagation.
5. Experiments and user studies demonstrate improved performance over prior work.

For claims (1), (2), (4), and (5), the evidence is generally clear and convincing. The qualitative results (e.g., Figures 2 and 6), quantitative metrics (Table 1), and user study outcomes (Figure 7) consistently support the claim that small box adjustments lead to better visual quality and control adherence without deviating much from the original inputs.

However, the claim regarding the non-differentiability of the existing attention formulation (claim (3)) is less convincing as currently presented. While the paper states that Equation (2) is non-differentiable due to discrete boundaries, this point is not explained in enough detail to be fully persuasive

**Requested Changes:**

### **Critical**
* **Strengthen the positioning vs. prior work.** The paper mentions several related methods in Section 2, but the experimental/storyline focus is mostly on Trailblazer/Peekaboo. It would really help to add a more explicit discussion of why some of the other works (already cited) aren’t directly comparable, or what specific advantage this “control input optimization” view brings compared to them. Even a short, direct paragraph would make the paper feel much more complete.

* **Clarify and better justify the non-differentiability claim.** The paper states the original attention edit (Equation (2)) is non-differentiable w.r.t. box parameters due to discrete mask boundaries, but the explanation isn’t fully convincing as written. I’d strongly recommend adding a clearer justification (e.g., explicitly pointing out the hard thresholding / step behavior in the binary mask and why that blocks or destabilizes gradients), so the motivation for the differentiable reformulation is rock-solid.

* **Fix missing/unclear definitions and setup details.** Some notation is introduced without definition (e.g., $\text{Sig}$ in Equation (5)). Also, the optimization procedure would benefit from being spelled out more concretely: which layers are included in the objective (one specific layer vs. multiple/all), and over which denoising steps the updates are applied. Making these details explicit will make the method easier to understand and reproduce.

### **Minor**
* **Citation / venue naming consistency.** Some venue names are formatted inconsistently (e.g., AAAI appearing both as the full organization name and as the conference proceedings name, and casing varies). It’d be good to clean this up so the bibliography looks consistent and polished.

---

> ### Author Response · Authors · 2026-01-30
> **Official Comment by Authors**
>
> We sincerely appreciate the reviewer for the comments and thoughtful feedback. We add our detailed responses below.
>
> **tqFk-A1: Comparison with other methods / Positioning vs prior work**
>
> Thank you for the comment. We additionally compare our method with FreeTraj, a training-free approach for controlled generation, and show that our method achieves higher human-preference scores and improved motion control. These results are included in **Table 3**.
>
> Other methods mentioned in the related work are not directly comparable in this setting. Motion-Zero does not release implementation code or evaluation data, making fair experimental comparison difficult. Ctrl-V requires task-specific training and conditions video generation on an **initial frame**, whereas our method is training-free and operates directly on text-to-video diffusion backbones. Generally, other methods also differ substantially in supervision regime or underlying model architecture, which limits fair comparison.
>
> Overall, our comparisons highlight that optimizing the control input itself provides consistent gains over established training-free baselines.
>
> **tqFk-A2: Clarification with Equation 2 and non-differentiability**
>
> Thank you for the valuable feedback. Trailblazer uses a binary mask M_*, and therefore provides no gradient with respect to its spatial positioning, introducing its core limitation. While the Gaussian mask M_G is differentiable with respect to the box coordinates, this differentiability is lost when M_* is applied. In contrast, our mask M_B is fully differentiable, enabling gradient-based optimization of the boxes.
>
> We included this source of non-differentiability in the revised paper to better motivate our differentiable reformulation.
>
> **tqFk-A3: Clarification on implementation details**
>
> Thank you for the detailed feedback.
>
> **Notation.** In Equation (5), Sig denotes the standard sigmoid function; we now explicitly define this in the main text.
>
>
> **Layers.** Due to space constraints, layer-level details were deferred to the supplementary material. Summarized in **Table 2**, we apply edits uniformly across all layers in the lower-channel blocks, following prior attention work.
>
> **Optimization procedure.** We run 40 denoising steps in total for Trailblazer backbone, and apply attention editing and box optimization during the first 5 steps. To improve clarity and reproducibility, we include Algorithm 1 in the revised paper, which explicitly details the optimization loop.
>
>
> **tqFk-A4: Citation and naming consistency**
>
> – Thank you for the detailed feedback. We included this in the revised paper.

---

> > ### Comment · Reviewer_tqFk · 2026-01-30
> >
> > Thank you for addressing most of my concerns. However, I am still confused about the differentiation issue, and I would appreciate one or more rounds of discussion with the authors to clarify it.
> >
> > As stated by the authors, once applying $M_*$, the gradients cannot be backpropagated. However, based on Equation (2), the gradient with respect to the coordinates appears to be directly computable as:
> >
> > $$ \frac{\partial A}{\partial I_{\text{coord}}} = \frac{\partial A}{\partial M_\mathcal{g}} \cdot \frac{\partial M_\mathcal{g}}{\partial I_{\text{coord}}} $$
> >
> > Since $I_{\text{coord}}$ can be written as $\big((l + r)/2,\ (t + b)/2\big)$, the gradient should further propagate to $(l, r, t, b)$, which denote the bounding-box coordinates.
> >
> > I therefore hope the authors can provide a more detailed explanation of this part, specifically clarifying why Equation (2) has a differentiability limitation and under what conditions gradients cannot be backpropagated.

---

> > > ### Author Response · Authors · 2026-02-02
> > > **Official Comment by Authors**
> > >
> > > We thank the reviewer for the thoughtful question. The first term in Eq. (2) does not provide a useful gradient for optimizing the box coordinates, since the hard mask $M_*$ represents a hard selection. As a result, the derivative of $M_*$ with respect to the box coordinates is zero almost everywhere and undefined at the mask boundaries. This limitation motivates prior work such as TUSK [a], which introduces relaxations of bounding-box representations.
> > >
> > > The second term may at first glance appear differentiable, but it is still evaluated under multiplication by $M_*$. Since Eq. (2) involves a product between $M_*$ and $M_{\mathcal g}$, applying the product rule yields one term involving the derivative of $M_*$, which vanishes almost everywhere, and another term involving derivatives of $M_{\mathcal g}$. While the latter term is non-zero, it only affects how values are distributed within the current box and provides limited gradient signal for moving or resizing the box itself.
> > >
> > > This is analogous to hard top-K selection: although gradients (or subgradients) may flow through the selected values, they do not provide a faithful signal for learning the selection boundaries without an explicit relaxation [b].
> > >
> > >
> > > [a] TUSK: Task-Agnostic Unsupervised Keypoints
> > >
> > > [b] Differentiable Top-K Operator with Optimal Transport

---

> > > > ### Comment · Reviewer_tqFk · 2026-02-04
> > > >
> > > > I really appreciate the discussion with the authors; it resolved my confusion, and I have already made my final decision.
> > > >
> > > > That said, I still encourage the authors to revise the paper for clarity. Based on the discussion, the core issue appears to be the use of an incorrect signal rather than non-differentiability, and the current presentation may lead to confusion on a first read. The explanation provided in this discussion thread also offers a clearer framing for comparing the proposed approach with existing methods, and incorporating it into the paper would strengthen the exposition.

---

> > > > > ### Author Response · Authors · 2026-02-04
> > > > > **Official Comment by Authors**
> > > > >
> > > > > Thank you for the comments and detailed feedback. We are glad we were able to resolve your confusion and would work towards incorporating the new insights from the discussions in this thread into the main paper. Thank you for highlighting the importance of our work and ways that further strengthen it.

---

### Review · Reviewer_pt2F · 2025-12-16

**Summary Of Contributions:**

The paper addresses the challenge of enforcing spatial control in text-to-video synthesis. The authors propose a training-free optimization method that adjusts user-provided bounding boxes (spatiotemporal tubelets) to better align with the video model's internal cross-attention maps. To achieve this, the authors introduce a differentiable smooth masking technique and an optimization objective that maximizes attention within the box while preserving background attention. The results demonstrate that these "minor adjustments" to the input coordinates can significantly improve video quality and subject fidelity without requiring model fine-tuning. A user study confirms preferences for this method over baselines like Trailblazer and Peekaboo.

**Audience:**

Yes

**Audience Explanation:**

Yes. The paper is relevant to researchers in generative media and controllable synthesis. The proposed "training-free" approach  and the insight that rigid adherence to control signals can degrade quality are valuable takeaways for the community. The finding that differentiable attention editing can replace heuristic masking is of particular interest to algorithm designers.

**Broader Impact Concerns:**

While the experiments focus on animals, the method improves the controllability of video generation generally. The authors should briefly acknowledge the standard implications of controllable media synthesis, such as the potential for creating deepfakes or misleading content involving humans, rather than stating there are no implications.

**Claims And Evidence:**

No

**Claims Explanation:**

The claims are partially supported. The paper successfully demonstrates that "optimizing" the box coordinates improves generation quality metrics (PickScore, HPSv2) and user preference compared to rigid control baselines.

However, the evidence regarding "user intent" is limited by the experimental design:

- **Dataset Limitation**: The experiments rely exclusively on trajectories extracted from the Animal Kingdom dataset using object detection (OWL-ViT). Automated detections are often jittery or noisy compared to genuine human inputs. It is unclear if the method’s success comes from "correcting" noisy dataset labels rather than adhering to deliberate user intent.

- **Lack of Diverse Trajectories**: Real-world user intent often involves smooth curves, stationary-to-moving transitions, or complex paths (e.g., zig-zag)[1]. The current evaluation filters out discontinuous trajectories, potentially ignoring the difficult cases where user control is most needed.

- **Missing Baselines for Complex Scenarios**: As noted in the Trailblazer project page, users often request complex compositions or morphing. The paper does not demonstrate how this optimization loop handles scenarios where the user strictly requires a specific path like compositing, morphing, and multiple[1].

[1]https://hohonu-vicml.github.io/Trailblazer.Page/

**Requested Changes:**

To strengthen the paper, I request the following additions:

- **Evaluation on Synthetic/Manual Trajectories**: Please include qualitative examples using manually drawn, smooth trajectories (similar to the original Trailblazer demos). This is crucial to prove the method works on intended motion, not just on filtered dataset detections.

- **Complex Movement Analysis**: Include examples of more challenging movements (e.g., a "U-turn" or zig-zag, similar to the original Trailblazer demos) to see if the optimization "smooths out" the user's intended complexity to satisfy the model, or if it retains the complex shape.

- **Clarification on "Adjustment" Limits**: Please add a discussion or ablation on how much the box is allowed to deviate. If the user draws a specific box and the optimization moves it significantly to improve quality, does this violate "user intent"? A visual comparison of the degree of drift for difficult prompts would be helpful.

---

> ### Author Response · Authors · 2026-01-30
> **Official Comment by Authors**
>
> We sincerely appreciate the reviewer for the comments and thoughtful feedback. We add our detailed responses below.
>
>
> **pt2F-A1: Boxes similar to genuine user intent**
>
> –	Thank you for the insightful comment. Evaluating controllable text-to-video models is inherently challenging, as generated videos lack ground-truth annotations. We attempt to provide a standard evaluation protocol using the Animal Kingdom dataset that enables large-scale comparison across methods.
> We agree that automated detections can be noisier than genuine user inputs. Which is why we included experiments with smoothly interpolated trajectories between start and end boxes in the original submission **(Supplementary Figure 12)**, which we believe better approximate deliberate user intent. In both cases, our method improves generation quality within the bounding box while maintaining adherence to the intended trajectory.
>
> **pt2F-A2: Lack of diverse or difficult scenarios**
>
> –	Thank you for the feedback. Our data setup already accounts for imperfect detections by interpolating missing boxes to obtain complete trajectories. Following your suggestion, we further include challenging scenarios, such as stationary-to-moving transitions, U-turns, and zig-zag paths, in **Figure 15** and **16**, and video results in the local website (_index_response.html_). These results show that our optimization preserves complex motion patterns.
>
>
> **pt2F-A3: Additional complex scenarios**
>
> –	Thank you for the comment. Our work focuses on single-object trajectory control, where we show that optimizing the control input itself improves generation quality. Multi-object composition and morphing involve additional challenges, including object interactions and competing constraints, and are beyond the scope of this paper.
> However, following your suggestion, we include qualitative morphing examples in **Figure 14**, demonstrating that our optimized boxes can also benefit such scenarios and may serve as a foundation for future multi-object extensions.
>
> **pt2F-A4: Clarification on adjustment limits**
>
> –	Thank you for the feedback. As shown in the original submission **(supplementary material Figure 11)**, we visually analyze the impact of box deviation penalties. A higher penalty (1.0) restricts movement and limits quality gains, while a very low penalty (0.001) allows excessive drift from user intent.
>
> We use a fixed penalty of 0.1 across all experiments, which balances adherence to user intent with improved generation quality.
>
> **pt2F-A5: Broader impact concerns**
>
> –	Thank you for the thoughtful comment. We agree that our method is applicable beyond animals, and generations can be abused with malicious intent. For example, controlling the movement of celebrities for personal gain. We discourage such practices and hope that providing proper awareness can alleviate or minimize variations of such intent.

---

> > ### Comment · Reviewer_pt2F · 2026-01-30
> > **All my concerns have been addressed**
> >
> > I have read the other reviews, authors' responses and the updated manuscript.
> > The authors have addressed all the concerns I had.

---

> > > ### Author Response · Authors · 2026-02-04
> > > **Official Comment by Authors**
> > >
> > > Thank you for the comments. We are glad, that we are able to address all your concerns.

---

### Review · Reviewer_mm2Q · 2026-01-15

**Summary Of Contributions:**

This paper proposes a training-free method to improve bounding-box–based control in text-to-video diffusion models by optimizing the user-provided bounding boxes themselves rather than modifying the model. The core idea is to slightly adjust bounding boxes so that they better align with internal attention maps of the diffusion model.

Technically, the method replaces the discrete binary masking used in prior work with a smooth, differentiable mask, enabling gradients with respect to bounding-box parameters. The authors then optimize the box locations using an objective that maximizes next-layer attention within the box, while enforcing a balancing term to preserve background attention and a regularizer to remain close to the original user input. The method is evaluated on two video diffusion backbones and compared primarily against Trailblazer and Peekaboo.

**Strengths.**

- The method is training-free and easy to integrate into existing attention-editing pipelines.

- Empirical results suggest that small box perturbations can affect perceived generation quality.

**Weaknesses.**

- The conceptual novelty over prior attention-guided control methods is limited.

- The paper lacks clarity on core algorithmic details and optimization semantics.

- Experimental evidence is insufficient to convincingly support the claimed mechanism.

- Visualization and ablation studies are weak relative to the central claims.

**Audience:**

No

**Audience Explanation:**

The idea that user-provided control signals may be misaligned with a model’s internal representations is not new, and similar themes have appeared in prior work on attention guidance and inference-time control. While the specific instantiation of optimizing bounding boxes may interest a narrow audience working directly on video layout control, the lack of conceptual depth and weak empirical grounding limit the broader appeal.

Without stronger theoretical insight, clearer algorithmic exposition, or more compelling experimental evidence, the paper’s contribution is incremental and unlikely to significantly influence future work.

**Broader Impact Concerns:**

This paper does not require a Broader Impact Statement, since it only leverages pretrained generative models and does not introduce additional ethical complications.

**Claims And Evidence:**

No

**Claims Explanation:**

The paper claims that (i) small adjustments to bounding boxes significantly improve generation quality, and (ii) this improvement arises from better alignment between user intent and internal attention dynamics. However, the provided evidence falls short in several key aspects:

**Unclear and under-specified optimization process.**

The paper does not clearly explain whether bounding boxes are optimized per frame or jointly across time, whether they are static or dynamic during sampling, or how optimization interleaves with diffusion steps. As a result, it is difficult to understand what is actually being optimized and when.

**Weak mechanistic validation.**

Although the method is motivated by attention alignment, the paper provides minimal visualization of attention evolution or box trajectories over optimization. There is no direct evidence that the proposed objective improves attention behavior in a meaningful or stable way, beyond indirect downstream effects.

**Limited ablations on critical design choices.**

Several central components—including the smoothing kernel design, the choice of layer at which attention is maximized, and the relative weighting of loss terms—are fixed heuristically with little justification. Without proper ablations, it is unclear whether the reported gains are robust or incidental.

**Baseline coverage is narrow.**

Comparisons are largely limited to Trailblazer and Peekaboo. The absence of stronger or more recent trajectory-control baselines weakens the empirical claims of superiority.

Overall, while the experiments suggest that box perturbations can matter, the paper does not convincingly demonstrate that the proposed method is principled, necessary, or broadly effective.

**Requested Changes:**

**Substantially clarify the method and optimization semantics.**

Precisely specify how bounding boxes are optimized across frames and timesteps, and how this interacts with the diffusion process.

**Provide direct evidence for the claimed attention-alignment mechanism.**

Include visualizations of attention maps and box trajectories throughout optimization to demonstrate that the method behaves as intended.

**Add comprehensive ablations of key components.**

This should include kernel choice, loss terms, loss weights, and layer selection. Without these, it is not possible to assess robustness.

**Strengthen baseline comparisons.**

Include additional, more recent trajectory or layout control methods, or clearly justify their exclusion.

**Other details**

Improve writing clarity and organization; the current presentation is difficult to follow and contributes to confusion about the method; evaluate on more challenging prompts and multi-object scenarios; discuss computational cost more critically, especially given the modest gains.

---

> ### Author Response · Authors · 2026-01-30
> **Official Comment by Authors**
>
> We sincerely appreciate the reviewer for the comments and thoughtful feedback. We add our detailed responses below.
>
> **mm2Q-A1: Clarification on the optimization process**
>
> –  Thank you for this comprehensive feedback. We optimize bounding boxes jointly across time, allowing the method to reason over the full trajectory rather than individual frames. We perform 40 denoising steps and apply attention editing and box optimization during the first 5 steps.
>
> To remove ambiguity, we include Algorithm 1 in the revised paper, which explicitly details how box optimization interleaves with the diffusion process.
>
>
> **mm2Q-A2: Mechanistic validation via attention visualization**
>
> –	Thanks for your insightful suggestion. To provide direct evidence for the attention-alignment mechanism, we include an interactive slider in index_response.html that visualizes how internal attention maps evolve during optimization. These visualizations show that our differentiable editing induces smooth, stable changes in attention behavior.
>
>
> **mm2Q-A3: Ablations on component design choices**
>
> Thank you for the comment. We include comprehensive ablations on key design choices in the supplementary material (Tables 4–7).
>
>
> Deviation penalty. Table 4 shows that excessive deviation leads to loss of user intent (lower mIoU), while overly restrictive penalties limit quality improvements. A penalty of 0.1 balances control and generation quality.
>
> Smoothing kernel normalization. Table 5 demonstrates that normalizing the smoothing kernel to produce consistent peak attention across layers significantly improves PickScore, HPSv2, and mIoU.
>
> Background preservation term. Table 6 confirms that the background attention term is necessary for maintaining scene coherence.
>
> Edge strength. Table 7 shows that very low edge strength behaves similarly to discontinuous masks (e.g., Trailblazer), degrading quality, while overly strong edges reduce control accuracy. Intermediate values (0.001–0.03) provide the best trade-off.
>
> Together, these ablations show that the reported gains are robust and not the result of incidental hyperparameter choices.
>
>
> **mm2Q-A4: Baseline coverage**
>
> –	Thank you for the comment. Trailblazer and Peekaboo are well-established baselines for box-based, training-free video control, making them appropriate reference points for our method. In addition, as discussed in **tqFk-A1**, other methods also differ substantially in supervision regime or underlying model architecture, which limits fair comparison. Our goal is not to replace these methods, but to demonstrate that optimizing the control input itself provides consistent gains within a widely used setting.
>
>
> **mm2Q-A5: Broad applicability**
>
> –	Thanks for your comment. Beyond bounding boxes, our work highlights a broader principle: control signals themselves are often misaligned with a model’s internal representations, and modest optimization can substantially improve outcomes.  As also mentioned by **Reviewer tqFk**, this perspective naturally extends to future work on optimizing other conditional inputs such as trajectories, sketches, or depth cues.
>
> **mm2Q-A6: Computational cost**
>
> –	Thanks for your comment. In terms of inference speed, our method is applicable to more recent video models, which can run faster, e.g., T2V Turbo. Our generations take one minute and 37 seconds, while Trailblazer + original T2V Turbo takes 48 seconds, both on an NVIDIA RTX A6000. While it increases inference time due to the additional optimization loop, our generations are significantly better, as shown in Figure 12.

---

> ### Comment · Reviewer_mm2Q · 2026-02-04
> **Official Comment by Reviewer mm2Q**
>
> Thank you for the detailed response. With the improved visualization, clarification, and detailed ablation studies included, the updated manuscripts is more sounding, and most of my concerns are addressed.
>
> On top of the method proposed, I now do believe that the paper poses valuable insights into resolving the misalignment between control signals and the model's internal representation. That said, it would greatly strengthen the paper if the authors can further highlight this claim.

---

> > ### Author Response · Authors · 2026-02-04
> > **Official Comment by Authors**
> >
> > Thank you for the comments. We are glad that the reviewer found our paper insights valuable. We will work towards further highlighting these insights in the paper.

---

### Decision · Action_Editor_zmJs · 2026-02-09

**Recommendation:** Accept as is

**Audience:**

Yes

**Audience Explanation:**

The paper addresses a topic that is of clear interest to the TMLR community.

**Claims And Evidence:**

Yes

**Claims Explanation:**

The reviewers are largely convinced that that the paper's claims are supported by the evidence. There's some concern that a few aspects of the claims could be explained or empirically demonstrated a bit better, or compared with more baselines. However, all things considered, the editor is inclined to give the authors leeway on this: the paper has demonstrated its claims sufficiently.